# Effects of Contact Pressure in Reflectance Photoplethysmography in an In Vitro Tissue-Vessel Phantom

**DOI:** 10.3390/s21248421

**Published:** 2021-12-16

**Authors:** James M. May, Elisa Mejía-Mejía, Michelle Nomoni, Karthik Budidha, Changmok Choi, Panicos A. Kyriacou

**Affiliations:** 1Research Centre for Biomedical Engineering, City, University of London, London EC1V 0HB, UK; Elisa.Mejia-Mejia@city.ac.uk (E.M.-M.); michelle.nomoni.1@city.ac.uk (M.N.); karthik.budidha@city.ac.uk (K.B.); p.kyriacou@city.ac.uk (P.A.K.); 2Samsung Advanced Institute for Technology, Seoul 16678, Korea; cm7.choi@samsung.com

**Keywords:** photoplethysmography (PPG), tissue phantoms, artificial blood vessels, contact pressure, signal to noise ratio, PPG features

## Abstract

With the continued development and rapid growth of wearable technologies, PPG has become increasingly common in everyday consumer devices such as smartphones and watches. There is, however, minimal knowledge on the effect of the contact pressure exerted by the sensor device on the PPG signal and how it might affect its morphology and the parameters being calculated. This study explores a controlled in vitro study to investigate the effect of continually applied contact pressure on PPG signals (signal-to-noise ratio (SNR) and 17 morphological PPG features) from an artificial tissue-vessel phantom across a range of simulated blood pressure values. This experiment confirmed that for reflectance PPG signal measurements for a given anatomical model, there exists an optimum sensor contact pressure (between 35.1 mmHg and 48.1 mmHg). Statistical analysis shows that temporal morphological features are less affected by contact pressure, lending credit to the hypothesis that for some physiological parameters, such as heart rate and respiration rate, the contact pressure of the sensor is of little significance, whereas the amplitude and geometric features can show significant change, and care must be taken when using morphological analysis for parameters such as SpO2 and assessing autonomic responses.

## 1. Introduction

Photoplethysmography (PPG) is an optical measurement technique used primarily for detecting volumetric changes of pulsatile blood flow in vascular tissue. It takes advantage of the fact that light is absorbed by body tissues, such as skin, fat, bone, and blood, in different amounts for any given wavelength of light. During the systolic phase of the heart, when blood is pumped from the pulmonary circulation to the systemic circulation via the left atrium and left ventricle of the heart, there is a momentary increase in the volume of blood in arteries, and the arteries expand. This increase in volume causes more light to be absorbed, resulting in less light being transmitted through the tissue. Utilising suitable sensors and instrumentation, these changes in light intensity can be detected and recorded, and the “pulses” detected can be used for anything from simple heart rate calculations to assessing cardiovascular health [1].

There are two modes of sensors used in PPG, transmission and reflectance. In transmission mode, the light source and detector are placed opposite one another inside a finger clip, and light is transmitted through the finger tissue. This is the most common mode seen mainly in pulse oximeters. The second type of mode is reflectance, where the source and detector are placed next to each other, and light is backscattered within the tissue. This mode is most commonly found in wearable devices, watches and armbands, or incorporated into mobile technology. Irrespective of the type of sensor used, the acquired PPG signal exhibits a quasi-periodic pattern consisting of an arterial pulse wave for each heartbeat. The morphology of this arterial pulse is known to be influenced by several physiological variations such as heart rate, heart rhythm, stroke volume, arterial stiffness, blood pressure, respiration, and the autonomic nervous system. This responsivity of the PPG signal to various physiological processes has helped researchers derive various diagnostic markers for vascular ageing and arterial compliance, large arterial stiffness, hypertension risk stratification, total peripheral resistance, atrial fibrillation, stress, endothelial dysfunction, gingivitis, apnoea, and autonomic nervous system (ANS) responses [2].

However, various other factors that are not physiological, such as sensor geometry, skin-sensor interface, contact pressure exerted by the sensor clip, photodiode sensitivity, ambient light, and measurement site, can also exhibit morphological changes in the PPG signal. In particular, the contact force or contact pressure (CP), the pressure exerted by a PPG sensor on the measurement location, is thought to influence the quality and morphology of the PPG signal significantly. Simple morphological features such as the systolic peak (used to calculate the heart rate), pulse width, pulse area, and the relative AC/DC amplitudes of red and infrared signals (used for oxygen saturation calculation) are thought to be adversely affected by an increase in sensor contract pressure [3]. In the past, several researchers have tried to investigate the effect of sensor contact pressure on the quality of the PPG signal. A summary of their findings, along with the type of sensor used and the type of force applied, is presented in Table 1. To make a likewise comparison between the studies and to underpin their findings, the contact pressure as described in the papers presented in Table 1, column 3 were converted to millimetres of mercury–mmHg and reported in Table 1, column 4. Every effort has been made to correctly identify the experimental setups of the reviewed studies so as to report the correct equivalent contact pressure. The force to pressure conversion was calculated using Equation (1):(1)PmmHg=FNewtonsSensor Actice Area m2133

As can be seen from the table, the effect of CP on the PPG signal acquired or the parameters estimated from the PPG signal is significant. Following these research paths still leaves a number of research questions viable for investigation. These are:How much is the degree of change in PPG signal features that an increase in sensor CP can create?What are the PPG signal features that are most affected?Is the change in a particular PPG signal feature significant enough to create misinterpretations when deriving diagnostic indices?

These questions and the question of the potential behaviour of arteries located under the PPG sensor when applied to wearable devices has not been investigated [2]. Other factors, such as the size of measurement location (e.g., finger diameter), have never been considered in any of the studies. Moreover, the in vivo studies reported thus far fail to isolate the sensor contact pressure at known blood pressure states specifically. This is obviously mostly due to the nature of recruiting volunteers or patients (who would naturally all vary widely in blood pressures due to physiological differences) and the known difficulty with trying to induce specific blood pressure states. Although methods such as the Valsalva Manoeuvre (VM) [10,11] have been used to induce lower pressures temporarily, this cannot be maintained indefinitely and is difficult to repeat successfully and concurrently to enable repeated measurements. 

All the above factors have led to a growing consensus that external factors that can affect the PPG signal quality and morphology, such as contact pressure, need to be further studied rigorously to help standardise PPG sensor design, which will, in turn, facilitate ideal PPG signal acquisition. This will also help standardise PPG measurements so that they are useful in not only standard monitoring (pulse oximetry and heart rate) but also for future and emerging applications such as Pulse Rate Variability (PRV) or Pulse Wave Velocity (PWV) measurements to assess cardiovascular health. 

As a first step towards discovering the optimal PPG sensor design and conditions, an in vitro investigation was carried out using a relatively simple vessel-tissue phantom with similar mechanical properties to human anatomy. The developed in vitro vessel-tissue phantom with a pulsatile fluidic flow was able to maintain and replicate various blood pressure states whilst under controlled and measurable compressive loads, and was simultaneously able to record PPG signals directly at the site of compression. Through this effort, we aim to determine what sensor-contact pressure is optimum to obtain PPG signals and explore which morphological PPG features are least or most affected by increasing contact pressure.

## 2. Materials and Methods

### 2.1. Pulsatile Pump and Tissue Phantom

A vessel-tissue phantom was constructed using a previously described method with a clear silicone elastomer (Sylgard™ 184, DOW, USA) with an altered catalyst ratio to better mimic the softness of human tissue [12]. The vessel size, embedded within the tissue phantom, was 24 mm (inner diameter) with a wall thickness of 200 µm. The surrounding tissue had dimensions of 60 × 12 × 10 mm (L × W × H) with the vessel set at a depth of 3 mm. The phantom was placed on a force-sensing plate on top of an amplified load cell with a 50 N rating (FC22 series, TE Connectivity, Switzerland). To supply the phantom with a continuous blood pressure-mimicking waveform, a linear-drive based pulsatile pump with a dedicated compliance vessel was used (BDC Laboratories, USA). A custom-made reflectance PPG sensor with a contact area of 1 cm^2^, incorporating an integrated optoelectronic sensor (SFH 7050 BIOFY, Osram, Germany), was placed atop the phantom, directly above the vessel; this was connected to a custom and dedicated PPG monitoring system [13,14]. The integrated sensor comprised three LEDs (660 nm, 525 nm, and 940 nm) and a photodiode with a central peak absorbance at 940 nm aligned longitudinally and held in place by a metal armature attached to a linear actuator able to travel in the orthogonal direction relative to the bed of the force-sensing plate (see Figure 1 and Figure 2). Only the red and infrared LEDs of the integrated sensor were utilised and operated in this design; hence, the final sensor is a dual-wavelength device. The pressure inside the phantom was monitored with a luer-connected inline blood pressure (IBP) probe at the inlet of the phantom (BDC Laboratories, USA).

### 2.2. Experiment Protocol

To investigate the effect of varying contact pressure on the PPG signal quality and how it may also affect the PPG SNR, morphology, and its features, a blood-mimicking fluid (distilled water and India ink solution) was continuously pulsed through the phantom via a branching circuit from a simplified human arterial network. The pulses were timed at 60 BPM (1 Hz) using a custom bell-pulse profile. Four blood pressure states were implemented by varying the amount of resistance on a clamp simulating total peripheral resistance (TPR). These pressures equated to the ranges for hypotensive, normotensive, stage 1 hypertensive, and stage 2 hypertensive conditions, and were confirmed by reading the pressure at the input of the phantom for 30 s prior to the main experiment. The systolic blood pressure (SBP), diastolic blood pressure (DPB), and mean arterial pressure (MAP) produced are reported for each state in Table 2.

On confirmation that the correct pressure-state had been established, the linear actuator descended the PPG sensor into the phantom at a rate of 0.03 mms^−1^, starting just above the phantom surface. The sensor was withdrawn once it had reached the depth of the vessel (3 mm), ensuring that the vessel had been completely occluded. PPG signals were recorded simultaneously with the load cell voltage and repeated 3 times.

### 2.3. Analysis Protocol

All signals were sampled at 1 kHz and were processed offline in MATLAB^®^ (The MathWorks, Natick, MA, USA), using a 4th order Infinite Impulse Response (IIR) low-pass zero-phase filter with a cut-off at 12 Hz. To preserve fine detail in the temporal domain for the PPG feature analysis, the signals were not resampled.

#### 2.3.1. PPG SNR Analysis

Unlike Electrocardiology, where the electrocardiogram (ECG) can be assessed quantifiably for quality, the PPG signal is highly dependent on the sensor technology used, the PPG signal bandwidth, and the sensor location monitored. As a result, there is no consensus on what constitutes adequate PPG signal quality, and instead, it has been discussed that assessing the PPG signal must be done separately for basic quality, then diagnostic quality [15]. Basic PPG quality can be determined through the identification of the main pulse peaks, whereas diagnostic quality would be where the signal has clearly defined features such as systolic and diastolic morphology. Because of this consideration, this analysis assumes that all noise filtered out using the IIR filter previously described is true noise and does not constitute any part of PPG feature identification from this study. The signal to noise ratio (SNR) is therefore calculated in the standard way for every second of each PPG signal using the inbuilt SNR function in MATLAB^®^.

#### 2.3.2. PPG Features Analysis

All signals obtained from the in vitro setup were segmented to analyse the portions of data in which the sensor was in contact with the tissue phantom and in which there was still pulsatile activity. These portions were automatically segmented applying the following process.

First, the signals were segmented from the location in which the load cell voltage was positive to the location of maximum contact pressure. Then, the envelope of the rectified PPG signal was obtained using a low-pass FIR filter with a cut-off frequency of 0.5 Hz and order of 5 times the sampling rate. The resulting envelope signal was also differentiated and then filtered using a low-pass FIR filter with 0.1 Hz cut-off frequency and order of 10 times the sampling rate. The initial point of the signal of interest was identified as the location in which the normalised first derivative of the envelope crossed the normalised envelope. Finally, the endpoint of the signal of interest was determined as the first onset of the envelope signal occurring after the minimum of the derivative of the envelope.

From the segmented signals, the cardiac cycles were automatically detected using the algorithm proposed by Li et al. [16]. Since it has been suggested that the point in which tangent lines from the maximum slope point and the valley of the pulse intersect (TI points) are more robust for certain PPG-based applications than other points, such as the systolic peaks [16,17], these points were identified and used for segmenting the independent cardiac cycles.

From each identified cardiac cycle, 17 PPG morphological features were extracted. These features are illustrated in Figure 3 and described in Table 3.

The aim of this analysis is to identify those morphological features of the PPG signal that are less affected by contact force. Therefore, the relationship between the contact force and the behaviour of each feature was assessed using Spearman correlation coefficients. By this analysis, the features with coefficients closer to zero will be those that are less affected by the sensor contact force, since a correlation coefficient closer to zero indicates a low correlation, and this would mean that the trend of the magnitude of a given feature does not have a similar trend to the changes in contact pressure.

## 3. Results

### 3.1. PPG SNR vs. Contact Pressure

In total, there were 24 recorded PPG signals (12 red, 12 infrared) with increasing contact force split into 3 repeated recordings for each blood pressure state. It can be observed in the example signal recording in Figure 4 that the recorded signal amplitude generally increases with increasing force up to a point where a maximum SNR value is recorded before the amplitude begins to decrease, as the vessel in the phantom is slowly becoming more restricted, until there is a sudden drop off in SNR when the vessel is completely occluded and the SNR becomes negative (signal noise is dominant). Table 4 reports the results of the SNR analysis, stating the optimal contact force recorded at maximum SNR values for each blood pressure state.

### 3.2. Effect of Contact Pressure on PPG Signal Features

Figure 5 depicts the results of the segmentation algorithm and the detection of the cycles from the segmented sections of the PPG signals, respectively. As can be seen, using the proposed methodology, it was possible to automatically detect and segment the pulsatile portion of the obtained signals. Before the green circle, the signal has a pulsatile component, but the sensor has not made contact with the vessel phantom yet, which explains the lower amplitude and low quality of this portion of the signal. After the red circle, the contact pressure is so high that the vessel is occluded, and there is no more fluid flowing through the vessel phantom.

Figure 6, Figure 7, Figure 8 and Figure 9 show the behaviour of the extracted features when compared to the contact force in hypotension, normotension, stage 1 and stage 2 hypertension, respectively, from PPG signals obtained using both red and infrared wavelengths. It can be observed that the red signals tend to show less variability among repeated runs and that time-related features are less affected by contact force. In these figures, the contact pressure is plotted against the magnitude of the extracted features. Those features that exhibit a linear behaviour, rather than a quasi-exponential one, are found to be less affected by contact pressure, i.e., the changes in contact pressure do not affect the magnitude of the feature. Also of interest, it can be observed that the behaviour of the extracted indices is not altered in a significant manner by the different blood pressure states, although some magnitudes differ in scale, especially those related to amplitude measurements.

### 3.3. Statistical Analysis

Figure 10 shows the correlation coefficients obtained after comparing the behaviour of the extracted PPG features and sensor contact force, both with red and infrared PPG signals. The absolute values of the correlation coefficients were organised in ascending order for each repeated measurement and each blood pressure state. These results are shown in Table 5 and Table 6.

In line with the observed behaviour of the features, it is observed here that the time-related features (highlighted in blue) are less correlated to contact force when compared to amplitude-related features (highlighted in yellow). Features F1 (cycle duration), F9 (x-coordinate of the centroid of the systolic phase of the pulse), F11 (x-coordinate of the centroid of the diastolic phase of the pulse), F13 (pulse width), and F14 (rise time) showed the lower correlations to contact force when measured from infrared signals, whereas F1, F11, F13, F14, and F15 (decay time) were the features with lower correlations to contact force when extracted from red PPG signals. There was no observed difference across or between the BP states either, showing that the blood pressure state in this setup did not play a role in either the observed time-related or amplitude-related features.

From Table 5 and Table 6, it can be seen that there are differences among replicas and among blood pressure states for the features with a lower correlation to contact pressure. This fact might be explained by the presence of noise, in the case of the comparison among replicas, and by the effect of blood pressure values, which affect features from the PPG in a different manner. Further statistical analyses, with more available data, should be performed in order to understand how significant these differences among blood pressure states are and how replicable this experiment is. Nonetheless, the fact that time-related indices cluster on the top of these tables, regardless of the replica, the blood pressure state, and the wavelength used (i.e., red or infrared), indicate that these features are more reliable when contact pressure is not controlled for.

## 4. Discussion

The quality and morphology of a PPG signal are known to be affected by several external factors. One such factor that is known to play a significant role is the sensor contact force or pressure. Several researchers including Teng and Zhang [4]; Grabovskis et al. [5]; Shimazaki et al. [6]; Kasbekar and Mendelson [7]; Lee et al. [8]; and Scardulla et al. [9] have previously tried to investigate the effect of contact pressure on the quality of the PPG signal and the basic morphological changes such as pulse amplitude. With more and more researchers now investigating the use of photoplethysmography (<550 peer-reviewed publications in 2020) for measuring parameters ranging from PRV to biometric recognition, it is important to understand the effect of sensor contact pressure on the morphological features of the PPG signal, particularly since very many of these measurement methods are based on detecting appropriate changes in the morphological features of the signal.

In an attempt to understand the above, in this work, we investigated the effect of sensor contact pressure on an in vitro vessel-tissue phantom which mimics the properties of human tissue. The phantom was built such that the PPG sensor can be placed directly (3 mm deep) above the vessel, facilitating the investigation of our research questions.

The nature of in vitro investigation, however, is limiting in that some interesting physiological phenomena and anatomical features are difficult or, at present, impossible to replicate. In favour of the simplified model presented, however, it does allow the investigation of simple external stimuli, such as contact pressure, without complicating the response with factors outside our control, such as vasodilation/contraction or age-related vessel compliance. The results of the in vitro experiment can therefore be used to further inform or compare results from future in vivo studies.

### 4.1. SNR Analysis

It has been shown in previous studies that there exists optimum sensor contact force/pressure and the reported physiological reading when compared to another or gold standard recordings [4,5,6,7,8,9]. In our experiment, the reported SNR values across the measurements show little variance among either the repeated experiments or between the blood pressure states, and an optimum sensor pressure is observed that is similar to the previous in vivo studies. This lends confidence to the in vitro method we have described, and with further development of our phantoms, a larger set of physiological parameters may be tested. It must be noted, however, that in the studies by Kasbekar and Mendelson, Shimazaki et al. and Scardulla et al. [6,7,9], an emphasis was mostly put on reducing motion artefact during exercise, whereas the study by Lee et al. [8], focused on the subtle respiratory variations on the PPG. Both of these scenarios (physical exercise and respiratory artefacts) were not simulated in our study.

The study by Grabvoskis et al. [5] was of particular interest to us, as the effects of CP on PPG morphology to assess arterial stiffness is an interesting development in the field of PPG for other physiological monitoring other than HR and SpO2, i.e., vessel compliance or arterial function monitoring. However, the study protocol did not allow us to make the same type of analysis as in Grabvoskis’ study, and further experiments would be needed to make a comparative analysis.

Whilst it is observed that the red PPG signals have lower SNR generally than the infrared signals, this can be explained by the fact that the infrared PPG monitoring in this phantom may be more robust either due to the sensor component geometry, the type of ink used or a combination of the two. It is also interesting to note that this relationship seems to reverse during the stage 2 hypertensive experiments and might be explained by some mechanical property of the phantom, though further analysis, and more rigorous experiments and a review of the phantom construction may be needed to confirm this. To test the hypothesis that this is related to a mechanical property, a new in vivo experimentation protocol should be developed to investigate whether the PPG signal can be used to directly infer a mechanical condition of anatomy, such as vessel compliance.

### 4.2. PPG Feature Analysis

It has been shown that the morphological features of the PPG are affected by the contact force induced by the sensor to the tissue [4,5,9]. The obtained results in this in vitro study show that the contact pressure affects primarily amplitude-related features, such as the area of the cycle or the pulse amplitude. On the contrary, time-based features, such as the duration of the cycles or the width of the cycle, are less affected by the contact force of the sensor, regardless of the blood pressure. Hence, applications that are mainly based on time-related features, such as pulse rate variability (PRV) analysis, could be performed regardless of the contact force applied by the sensor in the tissue, as long as the detection of the pulses does not rely on systolic peaks, as has been suggested in by Mejía-Mejía et al. [18]. However, in applications where amplitude features (pulse amplitude, area of the systolic peak and diastolic peak, Y-coordinate of the centroid of the systolic and diastolic pulses) are key to deriving diagnostic markers while measuring oxygen saturation in pulse oximetry, assessing vasoconstriction, vasodilation, venous function, measuring the ankle pressure, genital responses, blood pressure, and cardiac function, contact pressure plays a major role [19,20,21,22,23,24]. Physiologically, the PPG amplitude features are a result of a complex interaction of stroke volume, vascular compliance, and tissue congestion effects [25]. Hence non-optimal contact pressure could potentially result in incorrect assessments.

Interestingly, the variation among the repeated experiments for each measurement was less notorious for features extracted from red signals. This could have two different explanations; the red signals are less affected by noise in the in vitro setup and are more robust due to the type of dye used (India ink which is absorbed more by the red light than the infrared light in accordance with the complementary colour chart), or the features extracted from red PPG signals tend to be less affected by noise. Although it is not possible to certainly pinpoint the reason from this current study, further studies should aim to explain these differences, and they should take into account that the red features might be more repeatable in the face of different contact forces or in scenarios in which the contact force cannot be controlled. 

Furthermore, the behaviour of an arterial blood vessel located directly underneath the PPG sensor can be derived from this investigation. Physiologically, arteries have two important functions [26]. The first is a conduit function that allows blood to reach the periphery of the whole body; the second is a cushioning function that minimises sudden surges in the pressure of blood vessels by the stroke volume during a systolic cycle of the heart. For these purposes, arterial walls must be compliant enough to temporarily store a portion of the blood during systole and release it during diastole. Healthy arteries are highly distensible and show a nonlinear stress–strain response with an exponential stiffening effect at higher pressures [27]. This stiffening effect, common to all biological tissues, is based on the recruitment of the collagen fibrils, which are extremely stiff [28]. The arterial wall in the living body is pre-stretched, under a blood pressure load in the blood vessel; therefore, they are always in a stressed state. However, when an external pressure to the outer arterial wall is applied for any reason (such as when using a PPG sensor), the stress in the arterial wall decreases. When external pressure, for a particular case, is the same as the mean blood pressure in a vessel, the stress in the arterial wall becomes minimised. Since the exponential stiffening effect eventually disappears for this condition [29], the maximum volumetric change of the arterial wall by blood pulsation can be monitored by PPG [30]. This effect is perfectly exemplified by Figure 4. As can be seen from Figure 4, the amplitude of the PPG signal in Figure 4 increases with increasing sensor contact pressure up to a point where the external CP is similar to the internal pumping pressure, where a maximum SNR value is recorded before the amplitude begins to decrease, as the vessel in the phantom is slowly becoming more restricted, until there is a sudden drop off in SNR when the vessel is completely occluded and the SNR becomes negative (signal noise is dominant). The same effect repeated over several blood pressures should result in a positive pressure correction between the PPG oscillometry peak and the blood pressure inside the vessel. 

## 5. Conclusions

This study has shown conclusively that in an in vitro environment, where there is significant effort to mimic relevant human anatomy and physiological state, the pressure exerted by the PPG probe (approximately between 10 and about 50 mmHg) does not significantly affect the ability to detect and measure certain PPG morphological features. It has also shown that there does exist a point at which the pressure exerted by the PPG probe on tissue is optimal (between 35.1 mmHg and 48.1 mmHg); moreover, the pressures reported in this experiment are in line and of similar values to those already reported in the literature. Rigorous controlled in vitro experiments utilising customised tissue phantoms applied in the field of Photoplethysmography could pave the way in exploring further PPG related research.

## Figures and Tables

**Figure 1 sensors-21-08421-f001:**
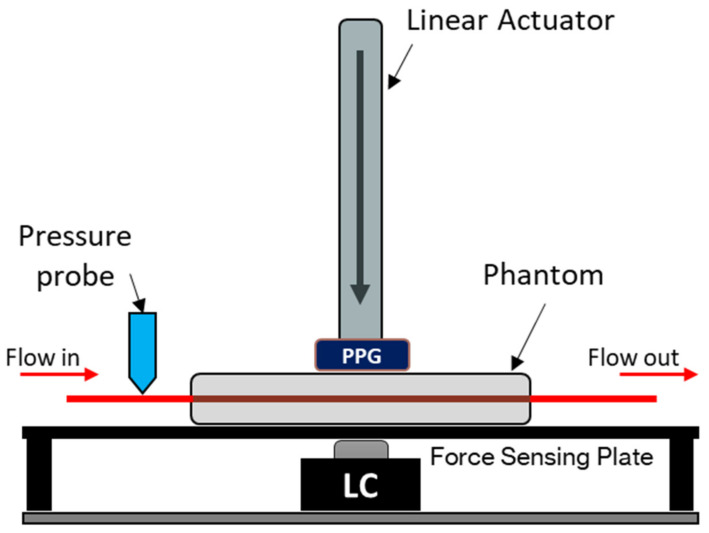
In vitro force-sensing setup. The PPG sensor (dual-wavelength, 660 nm and 940 nm) is mounted on the end of the linear actuator. The vascular tissue phantom rests on top of a force-sensing plate with a 50 N load cell (LC) as the force measuring device. A blood-mimicking fluid is pumped from the artificial vessel network, and the blood pressure is measured at the input of the phantom. The sensor position is measured from the surface of the phantom, where a positive increase indicates how much the sensor has descended into the phantom.

**Figure 2 sensors-21-08421-f002:**
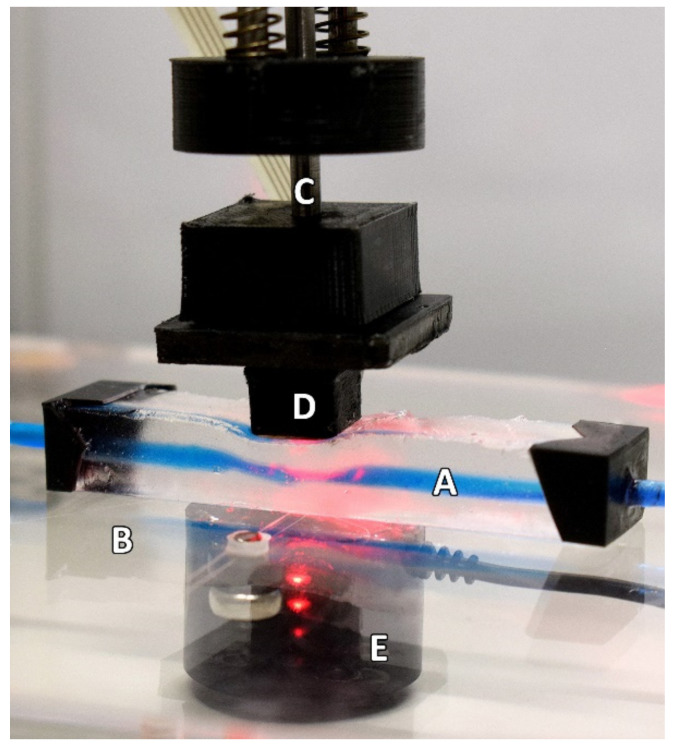
The experimental setup as seen during the continually applied pressure part of the protocol. The Phantom (**A**) is resting on a force-sensing plate (**B**) whilst the linear actuator (**C**) is descending. The PPG sensor (**D**) is seen resting on the phantom in this example during the descending phase of the protocol. The loadcell (**E**) can be seen under the force-sensing plate. For illustrative purposes, this depiction shows the use of a blue dye to highlight the artificial vessel better.

**Figure 3 sensors-21-08421-f003:**
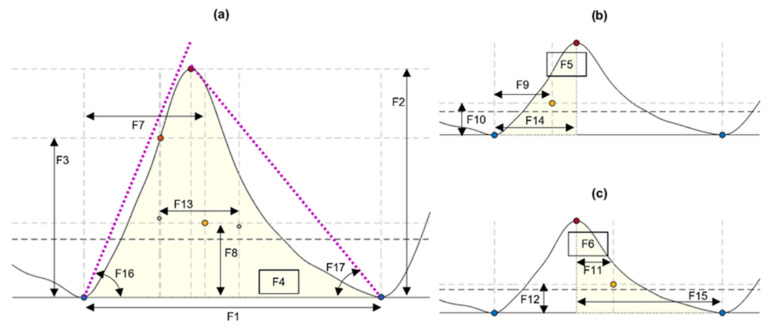
Features extracted from cardiac cycles of the photoplethysmograms, as described in Table 3. (**a**) Features extracted from the cardiac cycle. (**b**) Features extracted from the systolic phase of the cycle. (**c**) Features extracted from the diastolic phase. Blue circles: Onsets segmenting the pulse. Orange circles: Maximum slope point. Red circles: Systolic peak. Yellow circles: Centroid of each area of interest.

**Figure 4 sensors-21-08421-f004:**
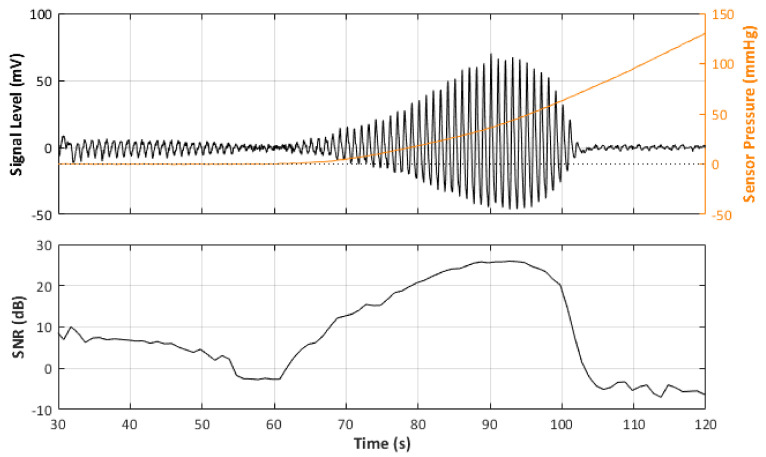
Signal Monitoring example from the experiment. Top plot shows the acquired PPG signal (black line) with the pressure at the probe (orange line). Bottom plot shows the calculated SNR during the pressure increase.

**Figure 5 sensors-21-08421-f005:**
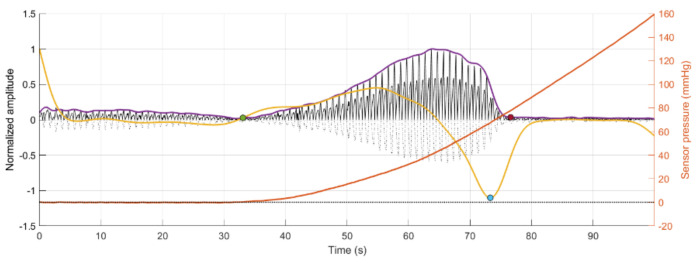
Segmentation of the area of interest for analysis, which is enclosed between the green and red circles, for each of the replicas of signals obtained from red (R) and infrared (IR) PPG signals. Dotted signal: Original PPG signal. Black signal: Rectified PPG signal. Purple line: Envelope of the rectified PPG signal. Yellow line: First derivative of the envelope signal. Blue circle: Minimum point of the first derivative signal. Orange line: Sensor pressure.

**Figure 6 sensors-21-08421-f006:**
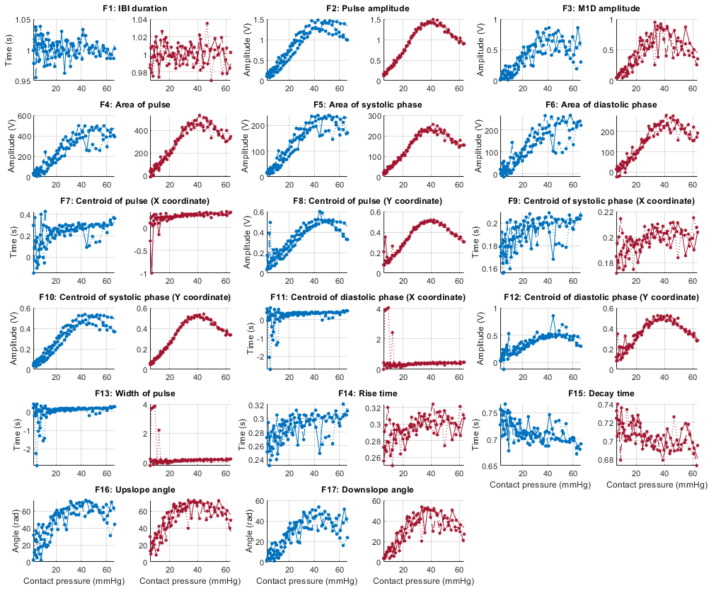
Behaviour of the extracted features during hypotension when compared against contact force. Blue lines: Infrared PPG-derived features. Red lines: Red PPG-derived features.

**Figure 7 sensors-21-08421-f007:**
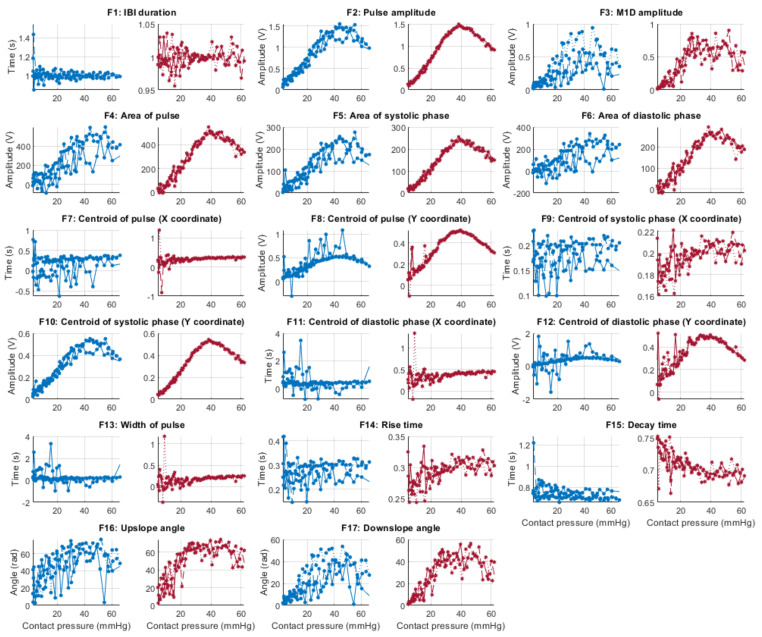
Behaviour of the extracted features during normotension when compared against contact force. Blue lines: Infrared PPG-derived features. Red lines: Red PPG-derived features.

**Figure 8 sensors-21-08421-f008:**
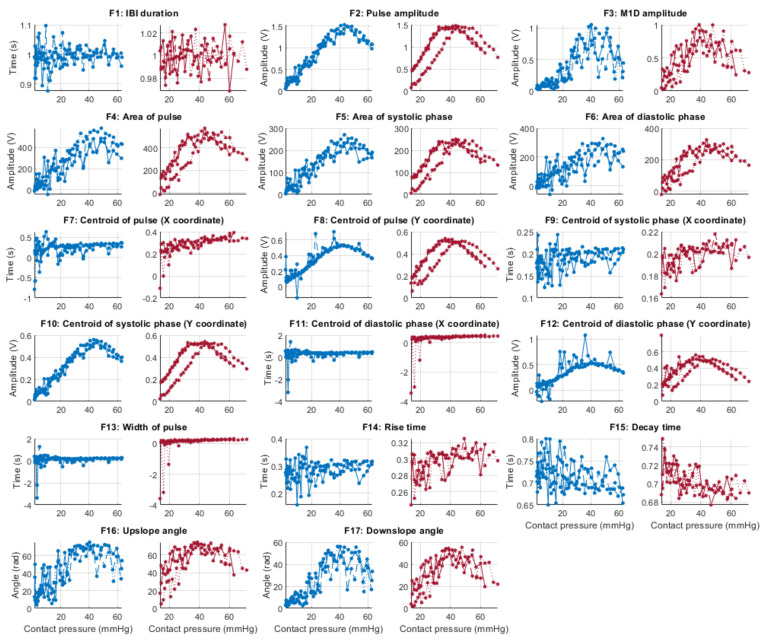
Behaviour of the extracted features during stage 1 hypertension when compared against contact force. Blue lines: Infrared PPG-derived features. Red lines: Red PPG-derived features.

**Figure 9 sensors-21-08421-f009:**
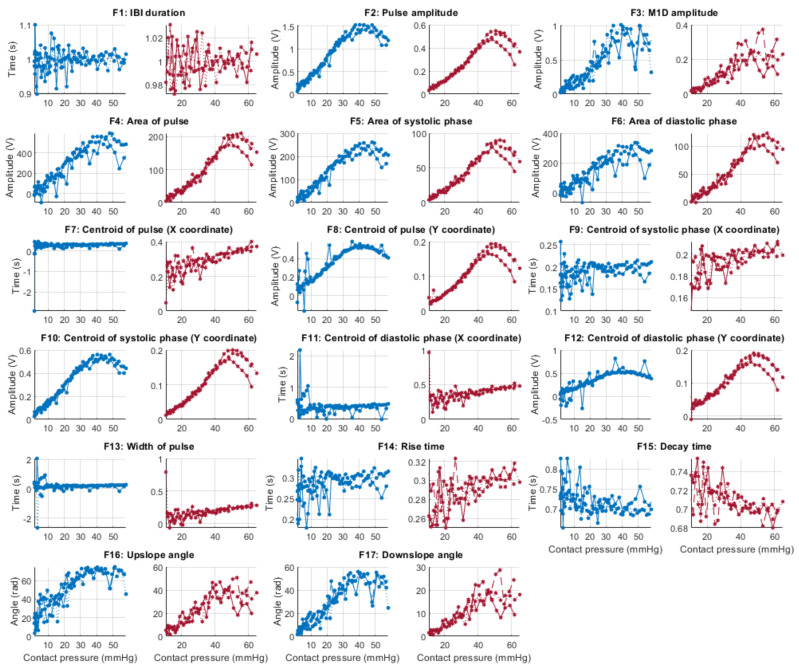
Behaviour of the extracted features during stage 2 hypertension when compared against contact force. Blue lines: Infrared PPG-derived features. Red lines: Red PPG-derived features.

**Figure 10 sensors-21-08421-f010:**
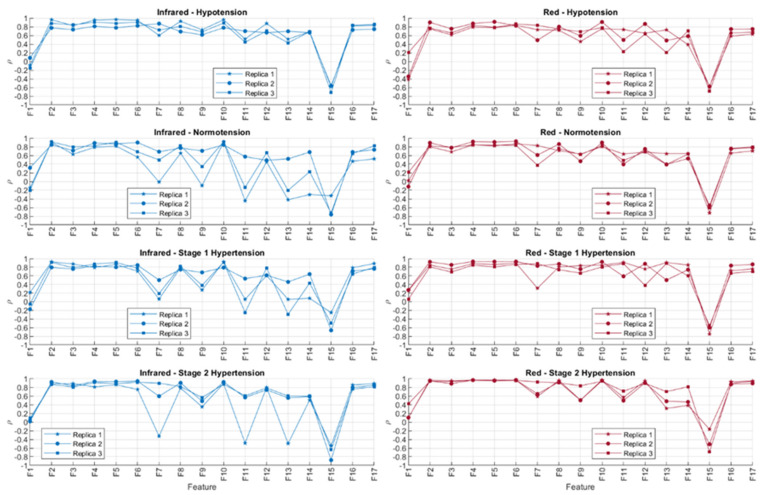
Correlation coefficients (ρ) obtained using Spearman correlation analysis for comparing features extracted from infrared and red photoplethysmograms and the contact force of the sensor. The experiment was run three independent times (replicas).

**Table 1 sensors-21-08421-t001:** Previous PPG force studies performed assessing sensor contact force to the quality of measurements made using PPG. ^A^ Assumption here is that the PPG probe was no larger than the force-sensing probe reported in the study; ^B^ Straight conversion kPa to mmHg (kPa times 7.501); ^C^ Straight Conversion hPa to mmhg (hPa divided by 1.333); ^D^ Same as assumption in ^A^, but also assuming researchers used the smallest version of the sensor reported available to account for the nature of the study (paediatrics, index finger). ^1^ [4]; ^2^ [5]; ^3^ [6]; ^4^ [7]; ^5^ [8]; ^6^ [9].

Study (1st Author)	Sensor Location	Optimum Pressure/Force Reported	Conversion to mmHg	Research Question	Findings
Teng ^1^	Finger	0.2–1.0 N	21–105 mmHg ^A^	Change in AC/DC ratio with change in contact force as an important metric when calculating blood oxygen saturation	PPG does have an optimum or “peak” value for the contact force applied. Hence careful sensor design consideration is required.
Grabovskis ^2^	Posterior Tibial A., Femoral A., Popliteal A.,	10.9, 11.8, 15.2 kPa	81, 88, 114 mmHg ^B^	The effect of probe contact pressure (CP) on the PPG signal for assessing arterial stiffness	Wrong contact pressure would adversely affect the AC PPG 2nd derivative peak ratio (known as the b/a ratio), a measurement index to assess arterial function. Also, suggests an optimal contact pressure.
Shimazaki ^3^	Forearm, Wrist	40–50 hPa at both locations	30–37 mmHg ^C^	The effect of fastening or applying contact pressure in wearable devices such as wristwatches which employ PPG to measure heart rate. Motion artefact reduction during exercise, elevated respiration artefact and accuracy of heart rate prediction were the key parameters investigated	All these studies reported that CP has a significant impact (i) reducing the noise introduced by motion artefact during exercise, (ii) increasing respiration related modulations in PPG, and (iii) increasing error in heart rate calculation up to ±11 beats per minute. Additionally, studies confirmed that further optimisation of the CP is indeed needed to reliably calculate physiological parameters.
Kasbekar ^4^	Forehead, Wrist	12 kPa (Forehead)	90 mmHg (Forehead) ^B^		
Lee ^5^	Index Finger (Paediatric Study, mean age = 4.1 y)	0.4–0.6 N	5.9–8.8 mmHg ^D^		
Scardulla ^6^	Wristband	54 mmHg	NA		

**Table 2 sensors-21-08421-t002:** Measured pressures in each blood pressure state at the input of the phantom.

	In Vitro Blood Pressure (mmHg)		
Blood Pressure State	SBP	DBP	MAP
Hypotensive	93	50	64
Normotensive	112	72	85
Stage 1 Hypertensive	143	104	117
Stage 2 Hypertensive	171	132	145

**Table 3 sensors-21-08421-t003:** Description of features extracted from the cardiac cycles detected from the photoplethysmograms.

Feature	Description (Units of Measurement)
F1	Cycle duration, measured from consecutive TI points (s)
F2	Pulse amplitude, measured as the amplitude difference between the onset of the pulse and its systolic peak (V)
F3	Amplitude of the maximum slope point, measured from the onset of the pulse (V)
F4	Area of the pulse, measured from consecutive TI points (V)
F5	Area of the systolic phase, measured from the onset of the pulse and the location of the systolic peak (V)
F6	Area of the diastolic phase, measured from the location of the systolic peak and the onset of the next pulse (V)
F7	x-coordinate of the centroid of the pulse (s)
F8	y-coordinate of the centroid of the pulse (V)
F9	x-coordinate of the centroid of the systolic phase of the pulse (s)
F10	y-coordinate of the centroid of the systolic phase of the pulse (V)
F11	x-coordinate of the centroid of the diastolic phase of the pulse (s)
F12	y-coordinate of the centroid of the diastolic phase of the pulse (V)
F13	Pulse width, measured as the difference between F9 and F11 (s)
F14	Rise time, measured as the time between the onset of the pulse and the location of the systolic peak (s)
F15	Decay time, measured as the time between the location of the systolic peak and the onset of the next pulse (s)
F16	Upslope angle, measured as the inverse tangent of the triangle formed by F2 and F14 (rad)
F17	Downslope angle, measured as the inverse tangent of the triangle formed by F2 and F15 (rad)

**Table 4 sensors-21-08421-t004:** Measured Maximum SNR and Pressures at each BP state with the standard deviations in parentheses.

	Hypotensive		Normotensive		Stage 1 Hypertensive		Stage 2 Hypertensive	
	SNR (dB)	Pressure (mmHg)	SNR (dB)	Pressure (mmHg)	SNR (dB)	Pressure (mmHg)	SNR (dB)	Pressure (mmHg)
**RED**	26.7 (<1)	38.6 (1.9)	26.0 (<1)	39.3 (<1)	25.9 (<1)	35.1 (2.0)	26.9 (<1)	48.2 (1.9)
**IR**	27.0 (<1)	44.1 (3.5)	27.3 (<1)	43.4 (1.7)	26.0 (<1)	39.0 (2.3)	25.8 (<1)	41.8 (1.8)

**Table 5 sensors-21-08421-t005:** Ordered features from the lowest |ρ| to the highest |ρ| for features (1 to 17) extracted from infrared PPG signals, in each of the runs of the experiment (R#) and under each blood pressure condition. Features coloured blue indicate temporal features, features coloured yellow indicate amplitude features, and those in green are geometric features reported in radians.

Hypotension			Normotension			Stage 1 Hypertension			Stage 2 Hypertension		
R1	R2	R3	R1	R2	R3	R1	R2	R3	R1	R2	R3
1	1	1	7	11	1	13	1	1	1	1	1
13	13	13	9	1	13	11	7	13	7	9	14
11	11	7	1	13	11	7	11	11	9	14	13
15	9	11	14	14	9	14	13	15	11	13	11
7	14	15	15	9	14	1	9	14	13	11	9
14	12	14	13	7	15	15	14	9	14	15	15
9	15	9	11	16	7	9	15	7	15	16	7
16	7	16	12	12	12	12	16	16	6	12	16
3	8	17	16	6	16	6	6	3	8	3	3
17	16	3	17	15	3	16	12	17	12	17	17
12	3	8	6	3	17	8	3	8	4	8	12
8	17	6	3	4	8	4	17	12	16	2	8
6	5	12	8	17	2	5	8	2	5	10	2
4	2	4	4	8	10	3	4	10	2	5	5
10	10	2	5	5	5	17	5	5	3	7	10
2	6	10	2	2	4	2	2	6	17	4	4
5	4	5	10	10	6	10	10	4	10	6	6

**Table 6 sensors-21-08421-t006:** Ordered features from the lowest |ρ| to the highest |ρ| for features (1 to 17) extracted from red PPG signals, in each of the runs of the experiment (R#) and under each blood pressure condition. Features coloured blue indicate temporal features, features coloured yellow indicate amplitude features, and those in green are geometric features reported in radians.

Hypotension			Normotension			Stage 1 Hypertension			Stage 2 Hypertension		
R1	R2	R3	R1	R2	R3	R1	R2	R3	R1	R2	R3
1	14	13	1	1	1	1	1	1	1	1	1
15	1	1	12	9	7	13	7	12	9	15	15
9	9	11	13	11	13	7	16	15	13	13	13
14	15	12	11	13	11	11	15	14	11	14	11
12	16	16	14	14	15	12	3	16	14	9	14
8	3	3	16	16	9	14	12	9	7	11	9
13	17	17	7	12	14	15	17	3	12	7	12
11	12	9	9	3	12	9	8	17	16	16	16
16	8	15	3	17	8	16	9	8	3	17	8
3	13	14	17	8	16	8	10	10	17	3	7
17	7	8	15	15	3	3	2	2	15	12	3
2	11	2	8	2	17	17	14	5	8	8	17
10	2	10	10	10	2	10	5	4	10	6	10
5	10	5	2	5	10	2	4	6	2	2	2
4	5	7	5	7	6	5	6	7	5	4	5
6	4	4	4	4	5	4	13	11	4	5	6
7	6	6	6	6	4	6	11	13	6	10	4

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
