# Peer review of "Effects of Contact Pressure in Reflectance Photoplethysmography in an In Vitro Tissue-Vessel Phantom"

_sensors, 2021, doi:10.3390/s21248421_

Round 1

Reviewer 1 Report

This study proposes an artificial tissue-vessel phantom to investigate the effect of applied contact pressure (CP) on PPG signals. The authors use signal-to-noise ratio and 17 morphological features to observe the effect of CP on PPG quality and morphology. They find that time-related but not amplitude-related features are more accurate in different blood pressure levels and CP values. Their findings generally are consistent with previous reports. In general, the manuscript is well-organized and shows good results and conclusions so that the readers will be interested.

MAJOR REVISION

  1. Limitation of the present study should be discussed in Discussion section.
  2. In Results section, both Figure 5 and Figures 6-9 deserve more description, respectively.

MINOR REVISION

  1. LINE 152

What does “be descending in thios replica” mean?

  1. LINEs 164-166

“The systolic blood pressure (SBP) diastolic blood pressure (DPB) and mean arterial pressure (MAP)…” should be “The systolic blood pressure (SBP), diastolic blood pressure (DPB) and mean arterial pressure (MAP)…”.

  1. LINE 175

” All signals were recorded at 1 kHz …” should be “All signals were sampled at 1 kHz …”.

  1. LINEs 227-228

Is “Those features with coefficients closer to zero can be considered as less affected by the sensor contact force.” correct?

  1. LINE 175

” All signals were recorded at 1 kHz …” should be “All signals were sampled at 1 kHz …”.

  1. LINE 244 (Table 4)

In Table 4, there are many parentheses. The numbers in those parentheses should be explained.

  1. LINE 318

”…morphological feathers…” should be “…morphological features …”.

Reviewer 2 Report

The paper investigates the effect of the contact pressure of Photoplethysmography (PPG) probes on the Signal to Noise (SNR) ratio of the PPG signal. The paper is well written and of great interest, however some concerns need to be addressed before publication:

  • The first concern is about the dimension of the vessel. In fact, it seems to simulate a large-caliber vessel. Could this aspect limit the generalization of the results only for large-caliber vessels?
  • As hypothesized by the Authors, the optimal contact pressure could increase with the blood pressure. However, the contact pressure corresponding to the highest SNR is higher for the normotensive with respect to stage 1 hypertensive. How the Authors explain this finding?
  • It is not clear to me if Figure 11 reports the correlation coefficient obtained between the PPG features and the contact pressure. A correlation coefficient of around 1 between the contact pressure and PPG features could indicate that the measure is highly affected by the pressure. Please better specify this aspect in the manuscript.
  • Tables 5 and 6 reports, accordingly with the caption, the features ordered from the lowest correlation coefficient to the highest. It seems that across the replicas and typology of patients (i.e. hypotension, normotension and hypertension stage 1 and 2) the feature with the highest correlation coefficient changes. How the Authors explain this aspect? It could be useful to report some statistics comparing the correlation coefficients, hence exploring statistical differences among them, among the replicas and the typology of patients.
  • Figure 9 is missing, please check. Moreover, from figure 6 to 10, please add the label at the x-axis.

Reviewer 3 Report

see attached file

Round 2

Reviewer 2 Report

The Authors replied to all my concerns and the paper is greatly improved. In my opionion the paper is suitable for publication in the present form.